

# Critical dynamics and cyclic memory retrieval
# in non-reciprocal Hopfield networks

**Shuyue Xue[1,2], Mohammad Maghrebi[1], George I. Mias[1,3,4] and Carlo Piermarocchi[1,2⋆]**

**1** Department of Physics and Astronomy, Michigan State University,
East Lansing, Michigan 48824, USA
**2** Department of Computational Mathematics, Science and Engineering,
Michigan State University, East Lansing, Michigan 48824, USA
**3** Institute for Quantitative Health Science and Engineering,
Michigan State University, East Lansing, Michigan 48824
**4** Department of Biochemistry and Molecular Biology,
Michigan State University East Lansing, Michigan 48824, USA

⋆ piermaro@msu.edu

## Abstract

We study Hopfield networks with non-reciprocal coupling inducing switches between memory patterns. Dynamical phase transitions occur between phases of no memory retrieval, retrieval of multiple point-attractors, and limit-cycles. The limit cycle phase is bounded by a Hopf bifurcation line and a fold bifurcation line. Autocorrelation scales as $\tilde{C}(\tau/N^\zeta)$, with $\zeta = 1/2$ on the Hopf line and $\zeta = 1/3$ on the fold line. Perturbations of strength $F$ on the Hopf line exhibit response times scaling as $|F|^{-2/3}$, while they induce switches in a controlled way within times scaling as $|F|^{-1/2}$ in the fold line. A Master Equation approach numerically verifies the critical behavior predicted analytically. We discuss how these networks could model biological processes near a critical threshold of cyclic instability evolving through multi-step transitions.

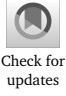

# 1   Introduction

The Hopfield model [1] is a spin glass model introduced to describe neural networks. It addresses the issue of content-addressable, or associative, memory, i.e., how some complex extended systems are able to recover a host of memories using only partial or noisy information. The statistical properties of Hopfield models have been extensively investigated (see, e.g., [2,3] for a review of earlier works). In typical Hopfield models, neural interactions are symmetric, but as Hopfield pointed out [1], the introduction of asymmetric interactions can result over time in transitions between memory patterns. In addition to the Hebbian coupling,

$$J_{ij} = \frac{1}{N} \sum_{\nu=1}^{p} \xi_i^\nu \xi_j^\nu, \tag{1}$$

where $\xi_i^\nu$, with $\nu = 1, \dots, p$ are spin memory patterns, one can introduce asymmetric interactions of the form

$$J'_{ij} = \frac{\lambda}{N} \sum_{\nu=1}^{q} \xi_i^{\nu+1} \xi_j^\nu, \tag{2}$$

with $q < p$. With this modification, some of the spin memory patterns become metastable and can be replaced in time by other patterns. This allows for the storage and retrieval of a limited number of temporal sequences of spin patterns. While incoherent asymmetry acts as a noise mechanism that can help stabilize memory retrieval [4], asymmetric interactions of the form in Eq. 2 enable coherent pattern evolution in time [5]. Moreover, the addition of terms of the form in Eq. 2 makes the spin system non-reciprocal. Recent studies have examined how non-reciprocity can induce novel classes of phase transitions that cannot be described using a free energy [6–8].

Dynamical spin models that can describe coherent temporal sequences, such as the class of Hopfield models above, are particularly interesting in the study of out-of-equilibrium processes. These models have recently been applied beyond modeling brain functions to many biological and biomedical systems, such as models of cell reprogramming [9, 10], classification of disease subtypes [11], or disease progression models [12, 13]. In particular, Szedlak et al. [14] used a Hopfield model with both terms in Eqs. 1 and 2 to describe the dynamics of gene expression patterns in the cell cycle of cancer and yeast cells. A key finding of the paper was the necessity of finely adjusting the model's parameters, specifically the noise level and the relative strength between symmetric and asymmetric interactions, embodied by the

parameter $\lambda$ in Eq. 2. This adjustment guarantees that the model maintains cyclic behavior while remaining sufficiently responsive to perturbations, such as targeted inhibitions that result in observable changes. This parameter tuning aligns with the idea of operating at the "edge of chaos", where biological systems exhibit both maximal robustness and sensitivity to external conditions [15].

Here, we study two-memory Hopfield networks with $N$-sites, characterized by asymmetric interactions that drive the system toward a critical threshold of oscillatory instability. The non-reciprocity leads to time-reversal symmetry breaking and introduces an extended region of criticality in the phase diagram, a feature typically observed in biological systems [16,17]. Similar behavior can be observed in other classes of non-reciprocal kinetic Ising models with on-site interactions between two different types of spin [18]. These asymmetric models can exhibit noise-induced interstate switching leading to non-equilibrium currents or oscillations [19]. Biological networks often operate far from the $N \to \infty$ limit. For instance, the cell cycle program only involves hundreds of genes. The role of fluctuations and their dependence on $N$ becomes, therefore, critical in their dynamical behavior. Here, we focus on the role of fluctuations in dynamical phase transitions to limit cycles. We find that the limit cycle phase is bounded by two critical lines: a Hopf bifurcation line and a fold bifurcation line. The autocorrelation function $C(\tau)$ on these lines scales as $C \sim \tilde{C}(\tau/N^\zeta)$, where $\tilde{C}$ are universal scale-invariant functions and $\zeta$ is a dynamical critical exponent previously introduced to characterize out-of-equilibrium critical behavior [20]. The dynamical exponent $\zeta = 1/2$ on the Hopf line and $\zeta = 1/3$ on the fold line. The sensitivity to an external perturbation of strength $F$ in these two critical regions also differs. On the Hopf line, the system exhibits enhanced sensitivity to periodic perturbations resonant with the limit cycle frequency and features a response time that scales as $|F|^{-2/3}$. In contrast, an external bias on the fold line can only induce switches between memory patterns in a limited and controlled way, without ever pushing the state into sustained limit cycles. Moreover, the characteristic response time is faster and scales as $|F|^{-1/2}$. While it was established that Hopf oscillators form a dynamic universality class relevant in biology, such as in the sensitivity of hair cells in the cochlea [21,22], the fold line identifies a distinct critical behavior that could help understanding transitions from stable points to cycles or more complex multi-step biological programs.

In Sect. 2, we introduce a two-memory non-reciprocal Hopfield model and analyze its phase diagram in a mean-field approximation. We show that the dynamical phase diagram is characterized by a cyclic behavior phase, bounded by two critical lines, Hopf and fold bifurcation lines. In Sect. 3 we examine the critical properties of the system on and near these two lines using analytical methods. We introduce an exact form of the Master Equation for the system in Sect. 4, explicitly accounting for the spin symmetry under pattern exchange. Based on this Master Equation approach, we explore the system in the large $N$ limit using a Glauber Monte Carlo procedure in Sect. 6. In Sect. 6.1, we numerically test the critical behavior and dynamic critical exponents derived analytically in Sect. 3. Finally, we summarize conclusions in Sect. 7.

## 2 Cyclic Hopfield networks

We consider Hopfield networks with $N$ Ising spins $\sigma_i = \pm 1$ interacting through non-reciprocal couplings $J_{ij} \neq J_{ji}$. We focus on a network encoding two memory patterns, $\xi_i^1$ and $\xi_i^2$, with couplings of the form

$$J_{ij} = \frac{\lambda_+}{N}\left(\xi_i^1\xi_j^1 + \xi_i^2\xi_j^2\right) + \frac{\lambda_-}{N}\left(\xi_i^1\xi_j^2 - \xi_i^2\xi_j^1\right). \tag{3}$$

The term proportional to $\lambda_+$ describes the Hebbian coupling, while $\lambda_-$ introduces a bias between the two memory patterns. By applying the Mattis gauge transformation [23] to the spins $\sigma_i \to \xi_i^1 \sigma_i$, $J_{ij}$ reduces to

$$J_{ij} = \frac{\lambda_+}{N}\left(1 + \xi_i \xi_j\right) + \frac{\lambda_-}{N}\left(\xi_j - \xi_i\right), \tag{4}$$

where $\xi_i = \xi_i^1 \xi_i^2$, which is equivalent to setting the first memory pattern to all spin up. The symmetric case, with $\lambda_- = 0$, has been previously introduced and solved by Van Hemmen [24]. In this two-memory Hopfield network, the $N$ spins separate into two sub-networks, which we call similarity ($S$) and differential ($D$) subnetworks [25]: $S$ corresponding to spins with $\xi_i = 1$ (i.e. $\xi_i^1 = \xi_i^2$) and $D$ corresponding to spins with $\xi_i = -1$ (i.e., $\xi_i^1 = -\xi_i^2$)). We can then define two magnetizations along the two memory patterns:

$$m_1 = \frac{1}{N}\sum_{j \in S,D} \sigma_j, \tag{5}$$

$$m_2 = \frac{1}{N}\sum_{j \in S}\sigma_j - \frac{1}{N}\sum_{j \in D}\sigma_j. \tag{6}$$

## 2.1 Mean field solution

To describe the dynamics of this Ising system we consider the probability $p_S(\sigma, t)$ that a spin in the subnetwork $S$ has spin $\sigma$ at time $t$. This probability satisfies in the large $N$ limit the equation [26, 27]

$$\frac{\partial p_S(\sigma, t)}{\partial t} = -p_S(\sigma)w_S(\sigma) + p_S(-\sigma)w_S(-\sigma), \tag{7}$$

where $w_S(\sigma)$ is the spin-flip transition rate. Assuming the system is in a thermal reservoir with inverse temperature $\beta = 1/k_B T$, the spin-flip transition rates take the form

$$w_S(\sigma) = (1 - \sigma \tanh \beta h_S)/2\tau_0, \tag{8}$$

where the field $h_S$ in the subnetwork $S$ and can be written as

$$h_S = (\lambda_+ - \lambda_-)\langle m_1 \rangle + (\lambda_+ + \lambda_-)\langle m_2 \rangle. \tag{9}$$

The $\langle m_{1(2)} \rangle$ are expectation values of Eqs. 5 and 6, and $\tau_0$ is an arbitrary constant that determines the time scale of Ising dynamics, originally introduced in one dimension by Glauber [28] and extended to higher dimensions by Suzuki and Kubo [29]. The equation for $p_D(\sigma, t)$, the probability for a spin $\sigma$ in the differential subnetwork $D$, is similar to the one in Eq. 7 but with a field:

$$h_D = (\lambda_+ + \lambda_-)\langle m_1 \rangle - (\lambda_+ - \lambda_-)\langle m_2 \rangle. \tag{10}$$

A mean-field system of equations for $\langle m_1 \rangle$ and $\langle m_2 \rangle$ can then be obtained from the equations for $p_S(\sigma, t)$ and $p_D(\sigma, t)$ as

$$\langle \dot{m_1} \rangle = -\frac{\langle m_1 \rangle}{\tau_0} + \frac{1}{\tau_0}\left\{n_S \tanh \beta \left[\lambda_a \langle m_1 \rangle + \lambda_s \langle m_2 \rangle\right] + n_D \tanh \beta \left[\lambda_s \langle m_1 \rangle - \lambda_a \langle m_2 \rangle\right]\right\}, \tag{11}$$

$$\langle \dot{m_2} \rangle = -\frac{\langle m_2 \rangle}{\tau_0} + \frac{1}{\tau_0}\left\{n_S \tanh \beta \left[\lambda_a \langle m_1 \rangle + \lambda_s \langle m_2 \rangle\right] - n_D \tanh \beta \left[\lambda_s \langle m_1 \rangle - \lambda_a \langle m_2 \rangle\right]\right\}, \tag{12}$$

where the coupling constants $\lambda_a = \lambda_+ - \lambda_-$ and $\lambda_s = \lambda_+ + \lambda_-$ account for the asymmetric and symmetric components of the interaction, respectively. The coefficients $n_{S(D)} = N_{S(D)}/N$ take into account the number of spins in the similarity and differential networks. Unless indicated,

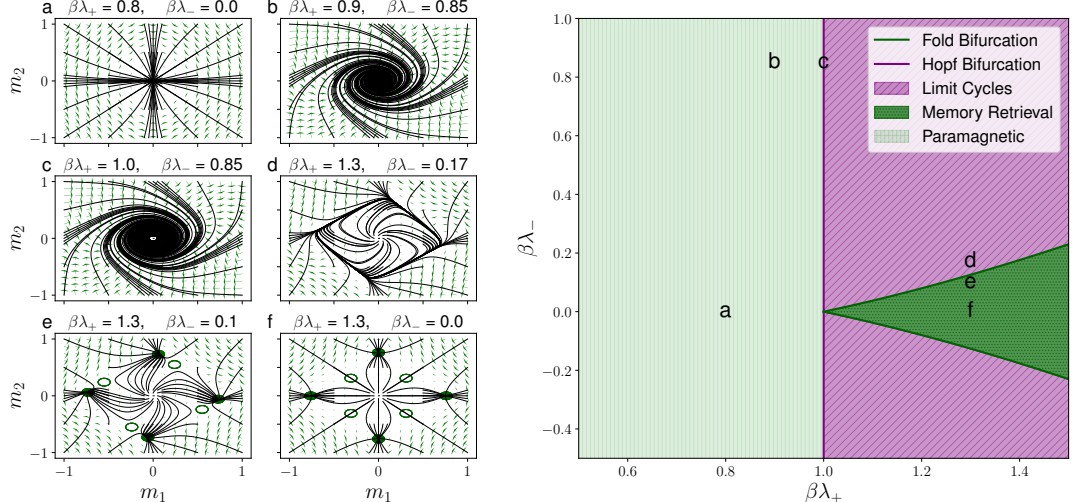

Figure 1: Phase portraits (left) and the phase diagram (right) for the two-memory cyclic Hopfield model. Left: The panel shows the dynamical behavior for different values of $\lambda_+$ and $\lambda_-$. The six phase portraits (**a** to **f**) show the trajectory dynamics, with green arrows indicating the vector fields of the derivatives. In **e** and **f**, empty circles represent saddle points, while solid circles denote stable points (sinks). The fixed point at $\mathbf{m} = 0$ is unstable (source) in **d**, **e**, and **f**. Right: The phase diagram is divided into three regions of different dynamical behavior: Limit Cycles (diagonal lines with a purple background), Memory Retrieval (dotted dark green background), and Paramagnetic (vertical stripes with a light green background). These phases are bounded by bifurcation lines: fold bifurcation lines (dark green lines) and the Hopf bifurcation (purple vertical line). The positions of the six trajectory plots (**a** - **f**) are indicated by corresponding labels on the phase diagram.

we will assume the case $N_D = N_S$ below. To simplify the notation, we drop the $\langle \ \rangle$ and assume mean field variables.

In Fig. 1, the phase portrait of the mean-field equations is presented. For $\lambda_- = 0$, a phase transition occurs at $\beta\lambda_+ = 1$. This transition separates the paramagnetic phase, (see orbits in **a**), from the memory retrieval phase (see orbits in **f**). In the latter phase, either $m_1 \neq 0$ or $m_2 \neq 0$, and $|m_{1(2)}| \to 1$ when $\beta\lambda_+ \gg 1$. Symmetric steady-state solutions with $m_1 = m_2 \neq 0 \leq 1/2$ are observed for $\beta\lambda_+ > 1$. These solutions, represented as empty circles in **f**, are mixed memory states equidistant from the two patterns. Such symmetric states are saddle point solutions and are always unstable in a two-memory scenario. In this reciprocal two-memory model, mixed asymmetric solutions are not permissible, in contrast to what is observed in Hopfield networks with more than two patterns, as shown by Amit et al. [30].

Next, we explore how these stable and unstable fixed points change in the presence of the asymmetric interaction $\lambda_-$. We rewrite Eqs. 11 and 12 in compact form as $\dot{\mathbf{m}} = F(\mathbf{m}, \lambda_+, \lambda_-)$, where $\mathbf{m}$ is the vector of magnetizations. In a neighborhood of $\overline{\mathbf{m}}$ which is a solution of $F(\mathbf{m}, \lambda_+, \lambda_-) = 0$, we can linearize the mean field equations as

$$\dot{\mathbf{m}} = \mathbf{A} \cdot (\mathbf{m} - \overline{\mathbf{m}}), \tag{13}$$

where the Jacobian matrix $\mathbf{A}$ at $\overline{\mathbf{m}}$ can be expressed as

$$\mathbf{A} = \begin{bmatrix} -1 + \beta\lambda_+\Delta + 2\beta\lambda_-\Gamma & \beta\lambda_-\Delta - 2\beta\lambda_+\Gamma \\ -\beta\lambda_-\Delta - 2\beta\lambda_+\Gamma & -1 + \beta\lambda_+\Delta - 2\beta\lambda_-\Gamma \end{bmatrix}, \tag{14}$$

with $\Delta = 1 - \overline{m_1}^2 - \overline{m_2}^2$ and $\Gamma = \overline{m_1}\,\overline{m_2}$.

The stability of steady-state solutions is determined by the eigenvalues of the Jacobian matrix $\mathbf{A}$. This matrix has either two real or two complex conjugate eigenvalues. In the scenario where $\overline{\mathbf{m}} = 0$, the eigenvalues of $\mathbf{A}$ are $\mu_\pm = \beta\lambda_+ - 1 \pm i\beta\lambda_-$, revealing that for $\beta\lambda_+ < 1$ and $\lambda_- \neq 0$, the solution $\overline{\mathbf{m}} = 0$ is stable, with focus-type orbits (see **b**). The line in $\beta\lambda_+ = 1$, where the eigenvalues transition to being purely imaginary is a Hopf bifurcation line. This line is the projection of the curve defined in $(m_1, m_2, \lambda_+, \lambda_-)$ by $F(\mathbf{m}, \lambda_+, \lambda_-) = 0$ and $Tr[\mathbf{A}(\mathbf{m}, \lambda_+, \lambda_-)] = 0$ on the $(\lambda_+, \lambda_-)$ plane [31].

When $\beta\lambda_+ > 1$, the phase diagram splits into two distinct regions, contingent upon the existence of solutions with $\overline{\mathbf{m}} \neq 0$. The boundary between these regions is a fold bifurcation line (also called saddle-node bifurcation line) derived from projecting the curve in $(m_1, m_2, \lambda_+, \lambda_-)$, defined by $F(\mathbf{m}, \lambda_+, \lambda_-) = 0$ and $Det[\mathbf{A}(\mathbf{m}, \lambda_+, \lambda_-)] = 0$ onto the $(\lambda_+, \lambda_-)$ plane [31]. Along this line, a single real eigenvalue transitions to zero while its counterpart maintains a negative value. This behavior can be interpreted as a merging of the memory retrieval fixed points, which are stable node-type, and the mixed memory states, which are saddle points. In this system, the non-reciprocal parameter $\lambda_-$ shifts the fixed points, causing four memory retrieval fixed points to approach the mixed memory states progressively. This convergence facilitates a circular directionality in the orbits, acting as a harbinger for the limit cycle solutions apparent above the fold bifurcation line, where only the unstable solution $\overline{\mathbf{m}} = 0$ persists.

## 2.2 Near cusp dynamics

The limit cycle phase can be better described by introducing a complex variable $z = m_1 - im_2$. By approximating $\tanh(x) \approx x - x^3/3$ in Eqs. 11 and 12, we obtain an equation for $z(t/\tau_0)$:

$$\dot{z} = (\Lambda - 1)z - \frac{\Lambda^2\bar{\Lambda}}{2}z^2\bar{z} + \frac{\bar{\Lambda}^3}{6}\bar{z}^3\,, \tag{15}$$

where $\Lambda = \beta(\lambda_+ + i\lambda_-)$, the bar indicates complex conjugation and the dot refers to derivation with respect to $t/\tau_0$. The last term in Eq. 15 is an anti-resonant term, which can be eliminated using a smooth change of variables:

$$w = z - \frac{h(\Lambda, \bar{\Lambda})}{6}\bar{z}^3\,, \tag{16}$$

where $h(\Lambda, \bar{\Lambda}) = \bar{\Lambda}^3/(3\bar{\Lambda} - \Lambda - 2)$. By substituting Eq. 16 into Eq. 15 and retaining only terms up to the cubic order in $w$, we obtain the Poincaré normal form:

$$\dot{w} = (\Lambda - 1)w - \frac{\Lambda^2\bar{\Lambda}}{2}w^2\bar{w}\,. \tag{17}$$

For $\rho(t) = |w(t)|$, we can then write:

$$\dot{\rho} = \rho\left[\beta\lambda_+ - 1 - \frac{(\beta\lambda_+)^2 + (\beta\lambda_-)^2}{2}(\beta\lambda_+)\rho^2\right]\,, \tag{18}$$

which indicates that non-zero steady solutions exist for $\beta\lambda_+ > 1$. Focusing near the cusp point at $\beta\lambda_+ = 1$ and $\beta\lambda_- = 0$ and retaining terms only up to the first order in $(\beta\lambda_+ - 1)$ and $\beta\lambda_-$, we find that the amplitude of the limit cycles increases with $\beta\lambda_+$ as:

$$\rho_0^2 = 2(\beta\lambda_+ - 1)\,. \tag{19}$$

To observe the change in the dynamical behavior corresponding to the fold line, we can write the equation for the phase $\theta(t) = \arg[z(t)]$ from Eq. 15, which keeps the anti-resonant term proportional to $\bar{z}^3$. Then, retaining terms up to the first order in $(\beta\lambda_+ - 1)$ and $\beta\lambda_-$ we have:

$$\dot{\theta} = \beta\lambda_- - \frac{\beta\lambda_+ - 1}{3}\sin 4\theta\,. \tag{20}$$

Using this equation, we can determine the period of the limit cycles as:

$$\frac{T}{\tau_0} = \frac{1}{4\beta\lambda_-}\int_0^{8\pi}\frac{d\theta}{1-\alpha\sin\theta} = \frac{1}{\beta\lambda_-}\int_0^{2\pi}\frac{d\theta}{1-\alpha\sin\theta} = \frac{2\pi}{\beta\lambda_-}\frac{1}{\sqrt{1-\alpha^2}}, \tag{21}$$

where $\alpha = (\beta\lambda_+ - 1)/3\beta\lambda_-$. Near the vertical Hopf line, the period is only determined by $\beta\lambda_-$. As we move right in the region with $\beta\lambda_+ > 1$, the period increases and then diverges when we approach the fold line, which, near the cusp point, corresponds to[1]

$$\beta\lambda_- = \frac{\beta\lambda_+ - 1}{3}. \tag{22}$$

## 3  Critical dynamics

To study the effect of fluctuations near the critical lines we modify Eq. 15 as

$$\dot{z} = (\Lambda - 1)z - \frac{|\Lambda|^2\Lambda}{2}|z|^2 z + \frac{\bar{\Lambda}^3}{6}\bar{z}^3 + \frac{1}{\sqrt{N}}\zeta(t), \tag{23}$$

where we have included the complex-valued white noise variable $\zeta(t)$,

$$\langle\zeta(t)\bar{\zeta}(t')\rangle = D\delta(t-t'), \tag{24}$$

to account for noise beyond the mean-field equation. In this section, we put $\tau_0 = 1$ to simplify the notation. The constant $D$ is phenomenological, and the scaling with the system size $N$ is chosen to match the standard mean-field equation plus noise for collective models (see, e.g., [20]). Let us consider the following distinct regions.

### 3.1  Near the cusp with $\beta\lambda_- = 0$ and $\beta\lambda_+ \approx 1$

In this case, the equation is better represented in terms of $m_1$ and $m_2$. In the absence of the asymmetric term, the dynamics is governed by an effective Hamiltonian $\mathcal{H}$ as

$$\dot{m}_i = -\frac{\partial\mathcal{H}}{\partial m_i} + \frac{1}{\sqrt{N}}\xi_i(t), \tag{25}$$

with a real white noise

$$\langle\xi_i(t)\xi_j(t')\rangle = D\delta_{ij}\delta(t-t'), \tag{26}$$

where

$$\mathcal{H} = \frac{r}{2}\left(m_1^2 + m_2^2\right) + u_1 m_1^4 + u_2 m_2^4 + 2u_{12}m_1^2 m_2^2. \tag{27}$$

Here $r = 1 - \beta\lambda_+$ and $u_1 = u_2 = \frac{(\beta\lambda_+)^3}{6} \approx \frac{1}{6}, u_{12} = \frac{(\beta\lambda_+)^3}{2} \approx \frac{1}{2}$. Note that the model exhibits a $Z_2 \times Z_2$ symmetry. The phase diagram is determined by the sign of $r$ and $u_1 u_2 - u_{12}^2$. Since $u_1 u_2 < u_{12}^2$, there are only three phases: $m_1 = m_2 = 0$ when $\beta\lambda_+ < 1$ and either $m_1 \neq 0 = m_2$

---

[1]The zero temperature $\beta = \infty$ limit of the Eqs. 11,12 can be studied by noting that $\tanh\beta x \to \text{sgn}\,x$ for $\beta \to \infty$. The only possible values for the steady states $m_1$ and $m_2$ are the ones compatible with the half sum of two sign functions, which can only give $0, \pm 1$ or $\pm 1/2$. The solutions with $|m_1| = 1$ and $|m_2| = 0$ and vice-versa describe the perfect memory retrieval. Consider the solution $m_1 = 1$ and $m_2 = 0$. By replacing these values in Eq. 11, we find $1 = (\text{sgn}\,\lambda_a + \text{sgn}\,\lambda_s)/2$ which is possible only for $\lambda_+ > \lambda_-$. If this condition is violated, the dynamics has limit cycles [27]. The equation for the fold line, which is given by Eq. 22 near the cusp, changes asymptotically to $\beta\lambda_- = \beta\lambda_+$ for $\beta$ approaching $\infty$. Also, the unstable mixed memory solutions with $|m_1| = |m_2| = 1/2$ exist in this limit only for $\lambda_- = 0$. This can be verified, for instance, by replacing the solution $m_1 = m_2 = 1/2$ in Eqs. 11 and 12, which give $\text{sgn}\,\lambda_+ + \text{sgn}\,\lambda_- = 1$ and $\text{sgn}\,\lambda_+ - \text{sgn}\,\lambda_- = 1$, and is possible only for $\lambda_- = 0$.

or $m_1 = 0 \neq m_2$ when $\beta\lambda_+ > 1$. This is analogous to a multi-critical point in a magnetic system where anisotropies break the $O_n$ rotational symmetry along more than one direction (see, e.g., Sect. 4.6 of Ref. [32]).

To understand the critical behavior at the critical point $\beta\lambda_+ = 1$ (and $\lambda_- = 0$), we consider the stochastic Langevin equation

$$\dot{m}_1 = -\left(4u_1 m_1^3 + 4u_{12} m_1 m_2^2\right) + \frac{1}{\sqrt{N}}\xi(t), \tag{28}$$

and a similar equation for $m_2$. The linear term vanishes since $r = 0$ at the critical point. Now, a rescaling of time and field variables,

$$\tilde{t} = t/N^{1/2}, \qquad \tilde{m}_i = N^{1/4} m_i, \tag{29}$$

leads to a scale-invariant equation (i.e., independent of $N$) as

$$d\tilde{m}_1/d\tilde{t} = -\left(4u_1 \tilde{m}_1^3 + 4u_{12}\tilde{m}_1\tilde{m}_2^2\right) + \xi(\tilde{t}), \tag{30}$$

and similarly for $m_2$. This observation leads to useful scaling relations. For example, the two-time correlation function for $t, \tau \gg \tau_0$ can be written as

$$C_{ij}(t,\tau) = \langle m_i(t+\tau)m_j(t)\rangle \sim \delta_{ij} N^{-1/2}\tilde{C}(\tau/\sqrt{N}), \tag{31}$$

where $\tilde{C}$ is a universal scaling function. The Kronecker delta function follows from the $Z_2 \times Z_2$ symmetry of the model.

## 3.2 Hopf bifurcation line: $\beta\lambda_- \neq 0$ while $\beta\lambda_+ = 1$

In this case, we use the transformation $w = e^{-i\beta\lambda_- t}z$, and we assume that the oscillation is sufficiently fast to neglect the anti-resonant terms, which can be viewed as a rotating wave approximation. The resulting equation for $w$ becomes

$$\dot{w} = -rw - \frac{1}{2}|w|^2 w + \frac{1}{\sqrt{N}}\zeta(t), \tag{32}$$

where we have replaced $\Lambda \approx 1$ in the nonlinear term. Note that the rotating wave approximation is equivalent to the Poincaré transformation in Eq. 16 for large $\beta\lambda_-$ and $\beta\lambda_+ = 1$. While the Poincaré method is more general and also valid in the small $\beta\lambda_-$ limit, we will discuss this case using the rotating wave Ansatz, which provides a more intuitive interpretation. Interestingly, the symmetry is now $O(2)$ rather than $Z_2 \times Z_2$. We can still describe the dynamics by an effective Hamiltonian defined as

$$\tilde{\mathcal{H}} = r|w|^2 + \frac{1}{4}|w|^4. \tag{33}$$

Similar approaches have appeared before [18, 33–35]. Scaling relations similar to the ones in Eq. 29 at the critical point $r = 0$

$$\tilde{t} = t/N^{1/2}, \qquad \tilde{w} = N^{1/4}w, \tag{34}$$

leads to a scale-invariant equation, and to

$$\langle w(t)\bar{w}(0)\rangle = N^{-1/2}\tilde{C}(t/\sqrt{N}). \tag{35}$$

The universal scaling function $\tilde{C}$ differs from the previous case because the underlying symmetries and the dynamics are different. We will show in Sect. 6.1 that this scaling behavior is consistent with numerical simulations.

## 3.3 Limit cycle phase near the Hopf line

The continuous $O(2)$ symmetry breaking in the ordered (limit cycle) phase in the regime where the rotating wave approximation applies results in a Goldstone mode, which is susceptible to noise. The interpretation of phase fluctuations in limit cycles as a Goldstone mode (see, e.g., Chapter 8 of Ref. [32]) has been used in recent works [33, 34]. Defining $w = \rho_0 e^{i\vartheta}$, the dynamics of the phase is given by

$$\dot{\vartheta} = \frac{1}{\rho_0 \sqrt{N}} \xi(t), \qquad \langle \xi(t)\xi(t') \rangle = D\delta(t-t'). \tag{36}$$

It then follows that $\langle (\vartheta(t)-\vartheta(0))^2 \rangle \sim Dt/(\rho_0^2 N)$. Therefore, the oscillations in the limit cycle phase are suppressed due to noise at any finite $N$ as (restoring $z = e^{i\beta\lambda_- t} w$)

$$C_z(t,\tau) = \langle z(t+\tau)\bar{z}(t) \rangle = \rho_0^2 e^{i\beta\lambda_-\tau - D\tau/(\rho_0^2 N)}, \tag{37}$$

thus oscillations remain coherent up to a characteristic time $T \sim \rho_0^2 N$. Below, we will show how the exact master equation approach and Glauber simulations reproduce this damping effect for finite systems. Deep in the limit cycle phase and/or closer to the fold line, the limit cycle dynamics is not uniform (i.e., not governed by a single frequency). However, we later show that the oscillations are similarly damped.

## 3.4 Fold line

Fluctuations on the fold line can be studied by adding a white noise term to Eq. 20. As one approaches the transition line $\beta\lambda_- = (\beta\lambda_+ - 1)/3$, the frequency of the limit cycle vanishes, and $\theta$ describes a soft mode. After making the transformation $\theta \to \theta + \pi/8$ and re-scaling noise strength and time using appropriate powers of $\beta\lambda_-$, we obtain the stochastic equation

$$\dot{\theta} = 1 - \cos(4\theta) + \frac{1}{\sqrt{N}} \xi(t). \tag{38}$$

Expanding around a fixed point, say $\theta = 0$, we find to the first nonzero order

$$\dot{\theta} \approx 8\theta^2 + \frac{1}{\sqrt{N}} \xi(t). \tag{39}$$

It follows from this equation that small but negative $\theta$ slowly converges to $\theta = 0$ while small but positive $\theta$ slowly diverges from $\theta = 0$ before a quick phase slip occurs from $0^+ \to \pi/2^-$. For an initial condition with $\theta(t=0) < 0$, the phase variable converges to $\theta = 0$ as

$$\theta(t) \sim -\frac{1}{8t}, \qquad t \to \infty. \tag{40}$$

The divergence for $\theta_0 = \theta(t=0) > 0$ is slow as well: the phase variable spends a time of the order $t \sim 2/\theta_0$ near $\theta = 0$ before a quick escape to a value close to, but below, $\theta = \pi/2$. Without noise, depending on the initial condition, the phase variable converges to one of the fixed points (in the above scenario, it would be $\theta = 0, \pi/2$). However, the noise qualitatively changes this picture.

As shown above, without nonlinearity, the noise will induce a mean square displacement given by $\langle (\theta(t)-\theta_0)^2 \rangle = 2Dt/N$. Therefore, even with $\theta_0 < 0$, noise would induce excursions to $\theta > 0$, followed by a long plateau, and then a quick slip to $\pi/2^-$ just below $\pi/2$. This is again followed by a noise-induced excursion to $\pi/2^+$ slightly above $\pi/2$, another long plateau, and then a phase slip to $\pi^-$, and so on and so forth; see Fig. 2. The resulting effect is a slow net rotation of the complex order parameter. Note that this rotation disappears as $N \to \infty$

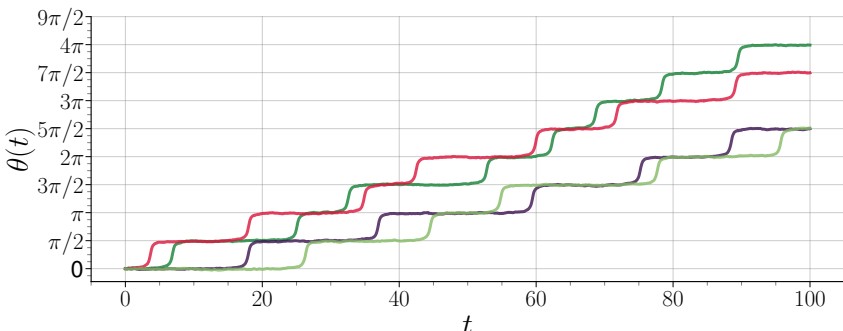

Figure 2: Representative trajectories at the fold transition as a function of $t$ with $N = 1300$ and $D = 1$. One can notice several features that are absent (deep) in the limit cycle phase: First, there is a larger variation between different trajectories. This highlights the role of noise in inducing phase slips. Second, the interval between jumps is rather long, and scales as $T \sim N^{1/3}$, in this case roughly of the order $T \sim 10$ compared to deep in the limit cycle phase where it is of order 1. Again this is due to noise as the period should diverge when $N \to \infty$.

since the noise is suppressed. The following argument gives the dynamical scaling behavior in $N$: Suppose we are close to $\theta = 0$. At short times, the nonlinearity is unimportant, while the noise induces a displacement of the order of $\theta(t)^2 \sim t/N$. At a sufficiently long time $t_*$, when $\theta$ is sufficiently large, and importantly also positive, the nonlinearity becomes relevant, making the phase variable diverge from $0^+$. A slow dynamics of the order $1/\theta(t_*)$ is followed by a quick phase slip before arriving at $\pi/2^-$. The time scale $t_*$ (or rather $\theta_* = \theta(t_*)$) is determined by minimizing (dropping constant factors for simplicity)

$$t_{\text{tot}} = N\theta_*^2 + \frac{1}{\theta_*}. \tag{41}$$

It follows that $\theta_* \sim N^{-1/3}$ and

$$t_* \sim N^{1/3}. \tag{42}$$

This means that the frequency of oscillations (at the critical point) goes to zero as $N^{-1/3}$. This behavior is also reflected in the correlation function, which for $t, \tau \gg \tau_0$ scales as

$$C_z(t, \tau) \sim \rho_0^2 \tilde{C}\left(\tau/N^{1/3}\right), \tag{43}$$

with $\tilde{C}$ being a scale-free function. This scaling behavior is verified, and the form of the scaling functions is calculated numerically in Sect. 6.1.

## 3.5 Near the fold line

Our discussion has focused on the phase transition exactly at the fold line. We next discuss what happens slightly away from this line into either the limit cycle or the fixed memory retrieval phase. In the limit cycle phase, another scale appears away from the phase transition, $\epsilon = \beta\lambda_- - (\beta\lambda_+ - 1)/3$ described by a modification of Eq. 38:

$$\dot{\theta} = 1 + \epsilon - \cos(4\theta) + \frac{1}{\sqrt{N}}\xi(t). \tag{44}$$

The motion is highly non-harmonic and resembles a step-wise rather than a smooth linear increase of the phase variable (hence, it is not described by a single frequency). Next, we

investigate whether the argument leading to Eq. 37 still follows and a damping with a characteristic time $T \sim N$ appears. The noiseless version of Eq. 44 admits an exact solution. While the precise form of the equation is not directly used in the following discussion, we report it for completeness:

$$\theta_0(t) = 2\tan^{-1}\left(\frac{\epsilon\tan\left(2\sqrt{\epsilon(\epsilon+2)}(t-t_0)\right)}{\sqrt{\epsilon(\epsilon+2)}}\right). \tag{45}$$

The characteristic oscillation frequency can then be extracted as $\omega_{\mathrm{LC}} \propto 2\sqrt{\epsilon(\epsilon+2)}$. To describe small fluctuations around this (noiseless) solution, we can take $t_0 \to -f(t)$ and expand the equation of motion to the first order in $f(t)$. Since $f(t) = $ const is an exact solution, the expansion only involves the time derivative, and we obtain

$$\frac{4\epsilon(\epsilon+2)}{\cos\left(4t\sqrt{\epsilon(\epsilon+2)}\right)+\epsilon+1}\dot{f} + \mathcal{O}(f^2) = \frac{1}{\sqrt{N}}\xi(t). \tag{46}$$

Also, we note that

$$\theta(t) = \theta_0(t) + \frac{4\epsilon(\epsilon+2)}{\cos\left(4t\sqrt{\epsilon(\epsilon+2)}\right)+\epsilon+1}f(t) + \mathcal{O}(f^2). \tag{47}$$

The same prefactor, which we denote by $\theta_1(t)$, appears in both equations above. We can then show that, up to a phase factor due to limit cycle oscillations, autocorrelation function scales as

$$|C_z(t,\tau)| \sim \rho_0^2 e^{-\frac{D}{2N}\theta_1^2(t)\int_t^{t+\tau}dt'1/\theta_1^2(t')}. \tag{48}$$

One can then see that the above term decays with time roughly exponentially (when coarse graining the features over each cycle) approximately as $\exp(-D\tau/2N)$, as in Eq. 37. We conclude that the latter equation is more general than the assumptions that were used to derive it, and is likely valid throughout the limit cycle phase. Indeed, the above equation suggests that the anharmonicity in the oscillations can be made uniform by reparametrizing the time as $d\tilde{t} = dt/\theta_1^2(t)$, leading to an equation similar to Eq. 36 that describes the dynamics of $\theta(\tau)$.

Near the fold line, the limit cycle frequency scales as $\omega_{\mathrm{LC}} \sim \sqrt{\epsilon}$. Comparing this with the behavior on the critical line, where $\omega \sim N^{-1/3}$, a rescaled variable $\epsilon N^{2/3}$ emerges governing the crossover between the two limits. This scaling follows from an application of the Arrhenius law on the other side of the phase transition where the point memory retrieval phase emerges. To this end, we consider $\epsilon < 0$ describing the point memory phase. We can approximate the dynamics by introducing a tilted Sine-Gordon effective potential (see Fig. 3):

$$V(\theta) = -(1-|\epsilon|)\theta + \frac{1}{4}\sin 4\theta. \tag{49}$$

For $\epsilon < 0$, a small barrier emerges, whose height scales[2] as $\Delta V \sim |\epsilon|^{3/2}$. Now according to the Arrhenius law, we find the decay rate given by $\Gamma \sim \exp(-\beta_{\mathrm{eff}}\Delta V)$, where the effective temperature, characterizing the noise strength, scales as $\beta_{\mathrm{eff}} \sim N$. Therefore, $\Gamma \sim \exp(-AN|\epsilon|^{3/2})$ and

$$C_z(t,\tau) \sim e^{-\Gamma\tau}. \tag{50}$$

Note that the same scaling variable ($|\epsilon|^{3/2}N$) governs both sides of the fold transition. This is a known characteristic dependence for fluctuation-induced transitions in nonlinear systems near bifurcations [36].

---

[2]More precisely, $V(-x_0) - V(x_0) \sim |\epsilon|^{3/2}$ where $V'(\pm x_0) = 0$.

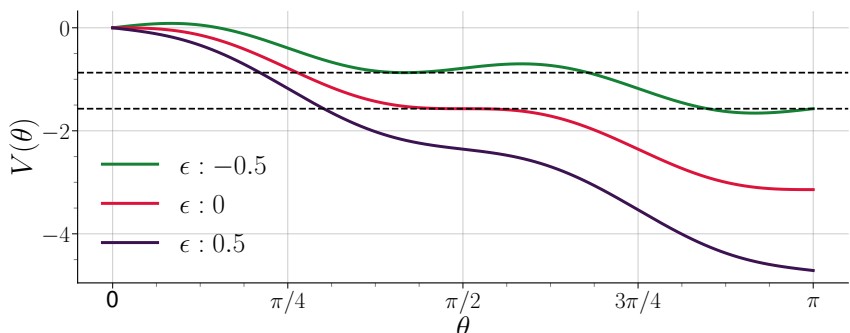

Figure 3: Effective potential $V(\theta)$ for different values of $\epsilon$. Depending on its sign, $\epsilon$ alters the steepness in $V(\theta)$, thereby affecting system's stability. A negative $\epsilon$ initiates an uphill start in $V$, which poses a potential hill for $\theta$ as a metastable state till it overcomes the hill. For $\epsilon = 0$, $V$ starts flat, then slips down at a faster rate than in the negative $\epsilon$ scenario. A positive $\epsilon$ triggers an immediate downhill movement in $V$, swiftly driving the system into the oscillatory phase at an even faster rate. The two horizontal dashed lines mark the $1^{st}$ local minima of the $\epsilon = -0.5$ and $\epsilon = 0$ cases.

## 3.6 External drive

Let us now consider the effect of an external drive on systems at the two critical lines. We assume that the external drive has the form $Fe^{i\omega t}$, where $\omega$ is nearly resonant with the cycle frequency, determined by $\lambda_-$ near the Hopf line, and approaching zero near the fold line. On the Hopf line, we can shift to a rotating frame by setting $w = e^{-i\omega_{LC}t}z$, which gives an equation similar to Eq. 32

$$\dot{w} = -i\delta w - \frac{1}{2}|w|^2 w + F, \tag{51}$$

with the detuning $\delta = \beta\lambda_- - \omega_{LC}$ while dropping the noise term. By rescaling to units $\tilde{t} = tF^{2/3}$ and $\tilde{w} = wF^{-1/3}$, Eq. 51 leads to

$$w(t) \propto F^{1/3}\tilde{w}(tF^{2/3}, \delta F^{-2/3}), \tag{52}$$

where $\tilde{w}$ is a parameter-free function. This suggests that, at $\delta = 0$, the response gain $w/F \sim F^{-2/3}$ diverges for small perturbations. Therefore, near the Hopf line, the system behaves like a filter with larger gain for weaker perturbations. This enhanced sensitivity at criticality is known to be relevant in biological functions, such as in the auditory sensitivity of hair cells in the cochlea [22]. The scaling analysis also shows that the dynamics at criticality is slowed down by a factor $F^{-2/3}$, so while smaller perturbations give enhanced gain, it also takes longer for the oscillator to respond to the external drive.

The response behavior on the fold line is qualitatively different. Since, without noise, the system is frozen on the fold line, we consider a constant complex drive $F$ with a fixed phase. Retaining only the phase dynamics, we find a modified Eq. 44

$$\dot{\theta} = 1 + \epsilon - \cos(4\theta) + \text{Im}\left(Fe^{-i\theta}\right). \tag{53}$$

The term $e^{-i\theta}$ can only be $\pm 1$ or $\pm i$ except during a fast switch between the memory pattern. Consider then a system initially frozen on the fold line (without noise) or in the memory retrieval phase with a small $\epsilon < 0$. The last term in Eq. 53 can shift the value of $\epsilon$ by $\pm\text{Re}F$ or $\pm\text{Im}F$, and a switch happens only if the result is positive. Since the sign of the shift is state-dependent, the maximum number of memory switches is limited to two. In other words, a static drive $F$ will never be able to push the system at criticality into a phase with sustained

limit cycles. Such a drive can only switch between memory patterns in a limited and controlled way. Finally, near an equilibrium position, a scaling analysis shows that

$$\theta(t) \propto F^{1/2}\Theta(tF^{1/2}, \epsilon/F), \tag{54}$$

at short times before the switch; here, $\Theta$ is a parameter-independent scaling function. This suggests that the response time on the fold line scales as $F^{-1/2}$ and is faster than the $F^{-2/3}$ dependence on the Hopf line.

## 4 Master equation

We now introduce a formulation for the Master Equation to describe the full dynamics of the network, allowing us to explore exactly the critical behavior studied in the previous section. Taking into account the separation of the full network into similarity and differential networks, we can rewrite the probability distribution at time $t$ for a given configuration of all spins $(\sigma_1, \sigma_2, \ldots, \sigma_N) = \{\sigma_i\}$ as:

$$P(\{\sigma_i\}, t) = \tilde{P}(M_S, M_D, t), \tag{55}$$

where the variables

$$M_{S(D)} \in \left[-N_{S(D)}, -N_{S(D)} + 2, \ldots, N_{S(D)}\right],$$

identify the sum of the spin configuration $\{\sigma_i\}$ over the subnetworks $S$ and $D$. Each value of $(M_S, M_D)$ is associated with a number of equivalent spin configurations given by:

$$g(M_S, M_D) = \binom{N_S}{N_{M_S}^+} * \binom{N_D}{N_{M_D}^+},$$

where $N_{M_{S(D)}}^{\pm} = (N_{S(D)} \pm M_{S(D)})/2$ indicate the number of spins up or down for a given $M_{S(D)}$. This degeneracy can be taken into account by defining a probability distribution

$$P(M_S, M_D, t) = g(M_S, M_D) * \tilde{P}(M_S, M_D, t), \tag{56}$$

which satisfies:

$$\sum_{M_S, M_D} P(M_S, M_D, t) = 1, \tag{57}$$

and its dynamics are determined by the Master Equation:

$$\frac{\partial P(M_S, M_D, t)}{\partial t} = I_{in} - I_{out}, \tag{58}$$

where

$$I_{in} = N_{M_S+2}^+ w_S^+(M_S + 2, M_D)P(M_S + 2, M_D, t) + N_{M_S-2}^- w_S^-(M_S - 2, M_D)P(M_S - 2, M_D, t)$$
$$+ N_{M_D+2}^+ w_D^+(M_S, M_D + 2)P(M_S, M_D + 2, t) + N_{M_D-2}^- w_D^-(M_S, M_D - 2)P(M_S, M_D - 2, t), \tag{59}$$
$$I_{out} = \left[N_{M_S}^+ w_S^+(M_S, M_D) + N_{M_S}^- w_S^-(M_S, M_D) + N_{M_D}^+ w_D^+(M_S, M_D) + N_{M_D}^- w_D^-(M_S, M_D)\right]P(M_S, M_D, t),$$

with $I_{in(out)}$ as the flux into (out of) the state $(M_S, M_D)$, and the $\pm$ spin-flip transition rates defined as

$$w_S^{\pm}(M_S, M_D) = \frac{1}{2\tau_0}\left(1 \mp \tanh\frac{2}{N}[\beta\lambda_+(M_S \mp 1) - \beta\lambda_- M_D]\right)$$
$$= \frac{1}{2\tau_0}\left(1 \mp \tanh\frac{2\beta}{N}h_S^{\pm}(M_S, M_D)\right), \tag{60}$$

$$w_D^\pm(M_S, M_D) = \frac{1}{2\tau_0}\left(1 \mp \tanh\frac{2}{N}\left[\beta\lambda_+(M_D \mp 1) + \beta\lambda_- M_S\right]\right)$$
$$= \frac{1}{2\tau_0}\left(1 \mp \tanh\frac{2\beta}{N}h_D^\pm(M_S, M_D)\right). \tag{61}$$

The terms $M_{S(D)} \mp 1$ in Eqs. 60 and 61 take into account the exclusion of the spin self-interaction. The effect of the $\pm 1/N$ is irrelevant in the mean-field solutions discussed above, and for the remainder of the paper, we focus on the case that omits self-interaction.

The single spin flip rates in Eqs. 60 and 61 can be rewritten in terms of the local energy change $\delta\epsilon_{S(D)}^\pm$ due to a spin-flip:

$$w_{S(D)}^\pm = \left(1 + e^{\beta\delta\epsilon_{S(D)}^\pm}\right)^{-1}, \tag{62}$$

where $\delta\epsilon_{S(D)}^\pm = -h_{S(D)}^\pm(M_S, M_D)\delta M_{S(D)}^\pm$ with $\delta M_{S(D)}^\pm = \mp 2$. Any cyclic process that starts from a given spin configuration and involves flipping only spins within either subnetwork $S$ or $D$ conserves the total energy, resulting in a net energy change of zero. However, when processes involve spins from both subnetworks $S$ and $D$, the energy change depends on the cycle path. Consider, for instance, the two-spin cycle

$$(M_S, M_D) \to (M_S - 2, M_D) \to (M_S - 2, M_D - 2)$$
$$\to (M_S, M_D - 2) \to (M_S, M_D), \tag{63}$$

where two spins up are sequentially flipped down and then back up, with the $S$ spin flipped before the $D$ spin. The total energy change in this case is $\delta\epsilon = -16\lambda_-/N$. In contrast, the time-reversed process in which the spin in $D$ is flipped before the one in $S$ results in $\delta\epsilon = +16\lambda_-/N$. This path dependence implies the violation of Kolmogorov's criterion for the transition rates [37, 38] and, therefore, breaking of the detailed balance principle.

## 5 Numerical diagonalization of Liouvillian

By enumerating the states using a single index $k = (M_S, M_D)$ we can rewrite the Master Equation in Eq. 58 as:

$$\dot{P}(k, t) = -\sum_{k'}\mathcal{L}_{k,k'}P(k', t), \tag{64}$$

where $\mathcal{L}$ is the Liouvillian matrix. The all-ones vector is always a left eigenvector of the non-symmetric matrix $\mathcal{L}$ with eigenvalue $\Lambda_1 = 0$, which guarantees the probability conservation in Eq. 57, and, for finite $N$, all the eigenvalues of the Liouvillian have a positive real part. To study the system's phase diagram, we focus in Fig. 4 on the second smallest eigenvalue $\Lambda_2$ and its dependence as a function of $N$. Note that the real part of $\Lambda_2$ remains nonzero in the region $\beta\lambda_+ < 1$ of the phase diagram in Fig. 1, corresponding to the paramagnetic phase. For $\beta\lambda_+ > 1$, the real part of $\Lambda_2$ converges to zero, allowing for the memory retrieval of a constant magnetization value as $N \to \infty$. The imaginary part of $\Lambda_2$, on the other hand, changes its behavior as a function of $N$ for $\beta\lambda_+ \sim 1.3$, which is near the fold line of the mean-field model, separating the limit cycle and the memory retrieval phases where the oscillations disappear. Observing the sharp features of the diagram in Fig. 1 by analyzing the eigenvalues of the Liouvillian is computationally demanding, and even for $N = 80$, resulting in an $\mathcal{L}$ of dimensions 1681 by 1681, the transitions in Fig. 4 are not sharply defined. Below, we will implement a Glauber Monte Carlo algorithm that allows us to explore significantly larger $N$.

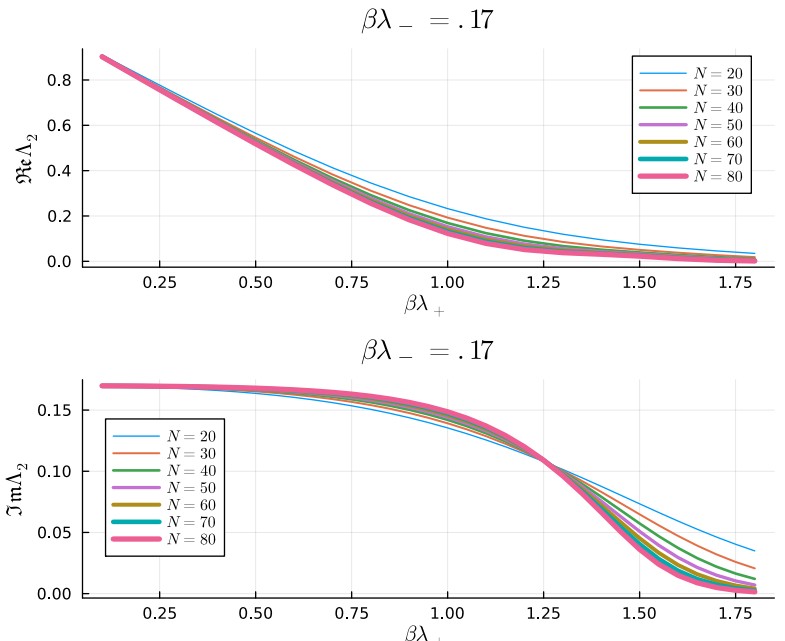

Figure 4: Real and Imaginary part of the second smallest eigenvalue of the Liouvillian matrix, $\Lambda_2$, as a function of $\beta\lambda_+$ for a fixed value of $\beta\lambda_- = 0.17$ and $N_S = N_D = N/2$.

## 5.1 Exact expectations and correlation functions

The Liouvillian matrix can be used to calculate exact expectation and correlation functions. For instance, given a probability distribution at $t = 0$, $P(k, 0)$, the average magnetization along the first memory pattern as a function of time can be calculated as

$$\langle m_1(t)\rangle = \frac{1}{N}\sum_k [M_1]_k P(k, t), \tag{65}$$

where

$$P(k, t) = \sum_{k'}\left[e^{-\mathcal{L}t}\right]_{k,k'} P(k', 0), \tag{66}$$

and $M_1 = M_S + M_D$. Similarly, the two-time correlation function for $M_1$ can be defined as

$$C_{1,1}(t, \tau) = \frac{1}{N^2}\sum_{k,\bar{k}} [M_1]_{\bar{k}} [M_1]_k P(\bar{k}, t+\tau; k, t). \tag{67}$$

The joint probability $P(\bar{k}, t+\tau; k, t)$ can be rewritten as

$$P(\bar{k}, t+\tau; k, t) = P(\bar{k}, t+\tau|k, t)P(k, t), \tag{68}$$

where $P(\bar{k}, t+\tau|k, t)$ is the conditional probability of the system to be in state $\bar{k}$ at time $t+\tau$, given it was in state $k$ at time $t$. This conditional probability can be calculated by shifting the initial condition $t \to 0$ and using

$$P(\bar{k}, t+\tau|k, t) = \sum_{k'}\left[e^{-\mathcal{L}\tau}\right]_{\bar{k},k'} P(k', 0), \tag{69}$$

with the initial probability set to $P(k', 0) = \delta_{k',k}$. The two-time correlation can then be expressed as [29]

$$C_{1,1}(t, \tau) = \frac{1}{N^2}\sum_k \langle M_1(\tau)\rangle_k [M_1]_k P(k, t), \tag{70}$$

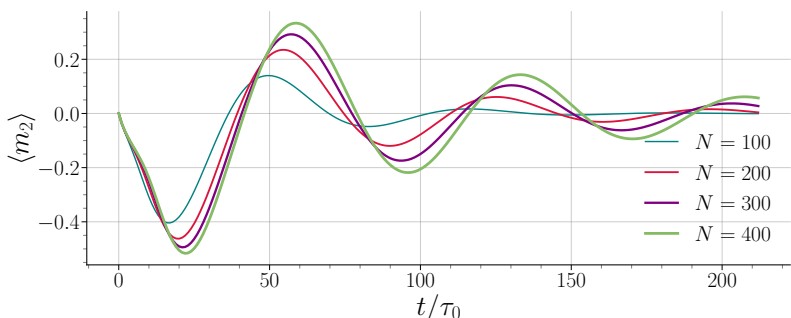

Figure 5: Exact $\langle m_2(t)\rangle$ solved from the master equation for different system sizes $N$ with $\beta\lambda_+ = 1.3$ and $\beta\lambda_- = 0.17$. As $N$ increases, the oscillations become slower and more pronounced.

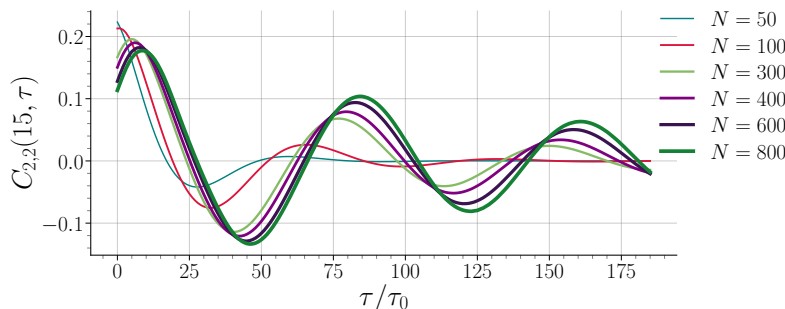

Figure 6: Two-time correlation function $C_{2,2}(t,\tau)$ for $M_2$ at $t = 15$, $\beta\lambda_+ = 1.3$ and $\beta\lambda_- = 0.17$, for various $N$. As $N$ increases, $C_{2,2}(t,\tau)$ exhibits defined oscillations.

where

$$\langle M_1(\tau)\rangle_k = \sum_{\bar{k}} [M_1]_{\bar{k}} \left[e^{-\mathcal{L}\tau}\right]_{\bar{k},k} \tag{71}$$

is the expectation of $M_1$ at $\tau$ given having been in configuration $k$ at $t = 0$. Similar averages and two-time correlations can be defined for other quantities such as $M_2$ and $Z = M_1 - iM_2$. Fig. 5 shows the exact $\langle m_2(t)\rangle$ for different values of $N$ calculated using the Liouvillian. The initial state was configured such that $m_1(0) = 1$ and $m_2(0) = 0$, with the parameters $\beta\lambda_+ = 1.3$ and $\beta\lambda_- = 0.17$. This positions the system slightly above the fold line in the phase diagram of Fig. 1. Fig. 6 shows the two-time correlation function for $M_2$, denoted as $C_{2,2}(t,\tau)$. The function is dependent on $N$ as well. In smaller systems ($N = 50, 100$), $C_{2,2}(t,\tau)$ quickly drops to zero, indicating that $M_2(t + \tau)$ becomes uncorrelated with $M_2(t)$ as $\tau$ increases due to fluctuations, while in larger systems, oscillations in $C_{2,2}(t,\tau)$ emerge.

## 6 Glauber dynamics

In parallel with deriving the master equations for $P(M_S, M_D, t)$, we implemented a Glauber dynamics that utilizes the division into subnets, rather than relying on random spin flips across the entire network. This adaptation not only provides a direct comparison with the predictions of the master equations but also allows us to examine much larger systems. Specifically, in our implementation, we consider the total magnetizations $M_1 = M_S + M_D$ and $M_2 = M_S - M_D$. Each Monte Carlo step involves a probabilistic decision to flip a spin within one of the two subnets, with the selection between $S$ and $D$ being randomized. The corresponding transition rates, as

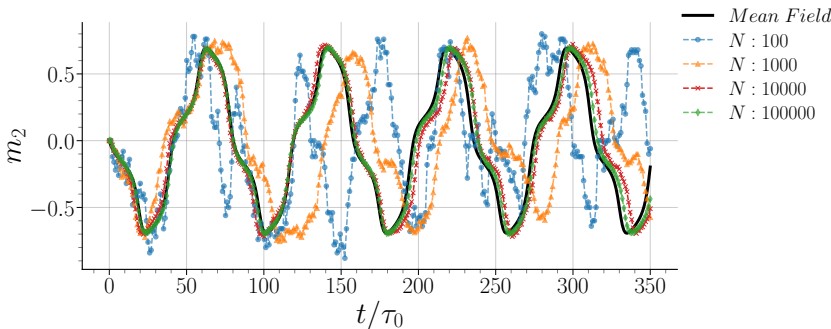

Figure 7: Magnetization $m_2(t)$ in Glauber dynamics as it approaches the mean field solution. Each dashed line with markers represents a single realization for system sizes $N = 100$ (blue circles), $1,000$ (orange triangles), $10,000$ (red crosses), and $100,000$ (green diamonds) at $\beta\lambda_+ = 1.3$ and $\beta\lambda_- = 0.17$, compared to the mean field solution (solid black line). As $N$ increases, the simulations match the mean field predictions, with larger systems nearly overlapping with the mean field curve.

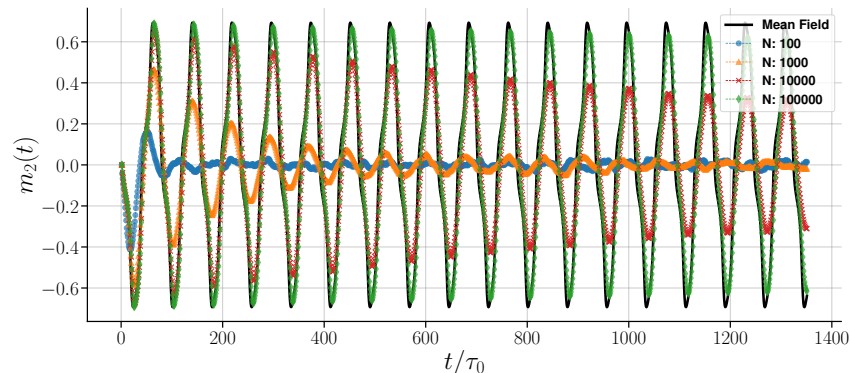

Figure 8: Average magnetization, $m_2(t)$ in Glauber dynamics over 1,000 realizations. The $N = 100$ system shows a significant decay within approximately two periods. At $N = 1,000$, the oscillation initially matches the mean field period but begins to shorten around $t/\tau_0 = 400$ while damping out. A larger system exhibits prolonged oscillation persistence, yet still with a noticeable damping. The largest N (the green curve) approximates an infinite system and more closely recovers the oscillations of the mean field solution.

defined in Eqs. 60 and 61, incorporate the effects of $\lambda_+$ and $\lambda_-$. Our implementation tracks these magnetizations at intervals of $N$ iterations. Below we show results from our simulations where we varied network sizes $N$, with additional adjustments in interaction strengths $\lambda_+$ and $\lambda_-$. We focused on assessing the system's finite-size effects and convergence towards the mean field solutions.

Fig. 7 illustrates the convergence of the Glauber dynamics toward the mean field solution for $N \to \infty$. The observations are consistent in both $m_1(t)$ and $m_2(t)$. At $N = 100$, deviations from the mean field solution are notable, particularly in the oscillation frequency and noise levels. As $N$ increases to $1,000$ and $10,000$, the discrepancies between the simulations and mean field solutions decrease, with progressively smoother magnetization dynamics. At $N = 10,000$ and $100,000$, stochastic effects significantly recede. In these larger systems, the dynamics closely resemble those of an infinite, continuous medium.

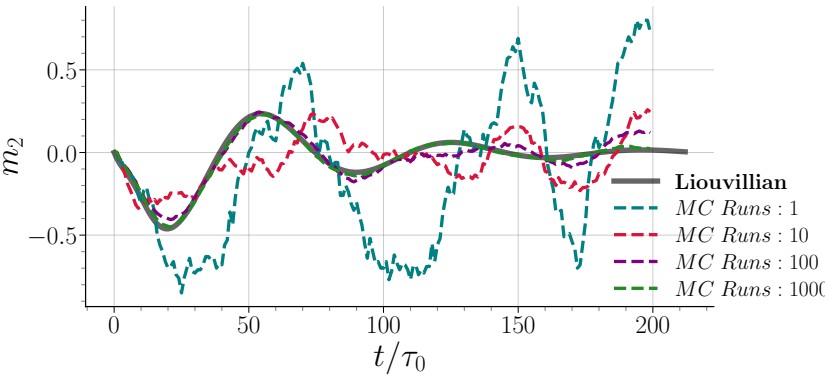

Figure 9: Equivalence of the master equation solution and averages of Glauber dynamics. System parameters and initial conditions are same of those in Fig. 7 and 8: $\beta\lambda_+ = 1.3$, $\beta\lambda_- = 0.17$, $m_1(0) = 1$, and $m_2(0) = 0$. The system has 100 spins in each subnet $S$ and $D$, totaling $N = 200$. Colored dashed curves represent the averages from Monte Carlo simulation runs. As the sampling size increases, the average trajectory of all stochastic paths converges to the Liouvillian solution.

While individual realizations of Glauber dynamics for very large $N$ align well with the mean-field solution, smaller systems exhibit significant variability. A comparison between ensemble-averaged simulations and the mean-field solution reveals damping as a net result of averaging over realizations. In Fig. 8, the averaged $m_2(t)$ displays oscillation damping, even in relatively large systems. This damping effect, due the ensemble average, remains pronounced even in a relatively larger system with $N = 1,000$. As expected, larger systems recover the mean field oscillation amplitude and maintain persistent oscillations over an extended range.

Fig. 9 contrasts the ensemble-averaged magnetization $m_2(t)$ from Glauber dynamics simulations with the exact Liouvillian solution of Fig. 5. As the sampling increases, the stochastic ensemble mean converges towards the Liouvillian dynamics, demonstrating the equivalence between the statistical expectations of stochastic processes and the deterministic predictions derived from the master equation. For larger sampling (purple and green trajectories), the averaging of individual dynamics leads to destructive interference and damped oscillations. A single realization (blue trajectory) still preserves the characteristic oscillation within the limit cycle regime, albeit with inconsistent periods. This can be attributed to the high susceptibility to noise in smaller systems. Our simulation was limited to $N = 200$, a relatively small configuration, due to the computational expense associated with the Liouvillian matrix calculation, as discussed above.

## 6.1 Numerical tests of critical behavior

In this last section, we test the predictions obtained using Langevin's equations in Sect. 3 with Glauber numerical simulations. We focus first on predictions related to the fold line. The first observation from Eq. 43 is that the oscillations of the autocorrelation function for a system exactly on the fold line are purely driven by fluctuations and are characterized by a period that scales as $N^{1/3}$. We show this behavior in Fig. 10, where after rescaling the delay time $\tau$ by $N^{1/3}$, the autocorrelation functions calculated numerically with $N$ ranging from $N = 1,000$ to $N = 50,000$ collapse to a single universal function. The autocorrelation is calculated starting at $t = 100\tau_0$ to remove transients related to the choice of the initial conditions. The expected scaling behavior is observed for the real and imaginary components of the autocorrelation of $z = m_1 - im_2$.

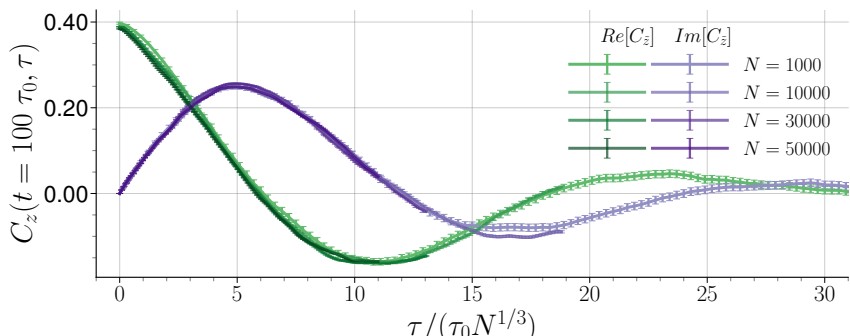

Figure 10: Autocorrelation on the fold line at $\beta\lambda_+ = 1.25$ and $\beta\lambda_- = 0.1025$. For different values of $N$. The time axis is scaled according to Eq. 43 to show collapsing into a single function. The absolute value of the autocorrelation decays purely exponentially, while its real and imaginary components show underdamped oscillations.

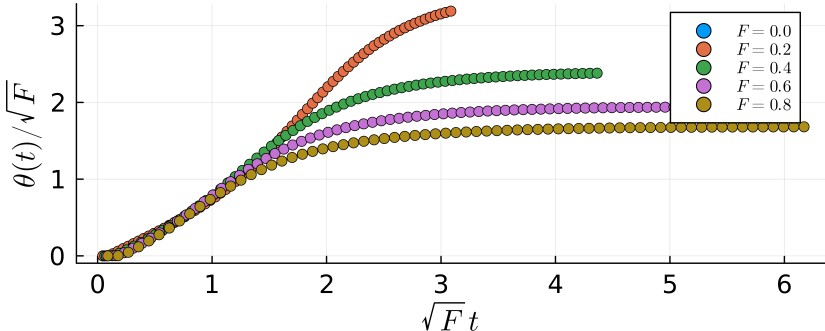

Figure 11: $\theta$ rotations following the activation of a constant $F$ for a system of $N = 10^6$ on the fold line with $\beta\lambda_+ = 1.25$ and $\beta\lambda_- = 0.1025$. Time and angles are rescaled according to Eq. 54, which is valid for $\theta \ll 1$.

The second prediction relates to the response of a system on the fold line to an external drive and its dependence on the strength of the drive, $F$. According to Eq. 54, we expect that in the limit of large $N$ where the noise-induced switching is suppressed, the characteristic time for switching scales as $F^{-1/2}$. We tested this behavior in Fig. 11, where we show the rotation of the angle $\theta = \arctan m_2/m_1$ right after the activation of a constant field $F$ in a system initially at $m_2 = 1$. The constant $F$ pushes the state towards $m_1$, and the amplitude of rotation and its time dependence scale as predicted by Eq. 54 in the limit of small $\theta$.

We also find $N$-dependent damped oscillations for the autocorrelation function on the Hopf bifurcation line. This is consistent with the scaling relation obtained in Eq. 35 using a rotating wave $\tilde{z} = e^{-i\beta\lambda_- t}z$. Fig. 12 shows how, by rescaling the autocorrelation in amplitude and time, simulation runs for networks of different sizes $N$ collapse into a universal function. We have verified that this behavior holds for different values of $\beta\lambda_-$ along the Hopf line.

Finally, we numerically studied the behavior of the autocorrelation functions slightly outside the critical lines, identifying two distinct behaviors. Near the Hopf line and above the fold line, the damping of the autocorrelation is associated with a characteristic time $T$ that scales linearly with $N$, following the analytical predictions of Eqs. 37 and 48. In contrast, below the fold line, in the regime where memory retrieval is effective, the characteristic time increases exponentially with $N$, as described by Eq. 50. Fig. 13 presents the results of numerical simulations where the decay of the autocorrelation function was fitted to an exponential model with a characteristic time $T$.

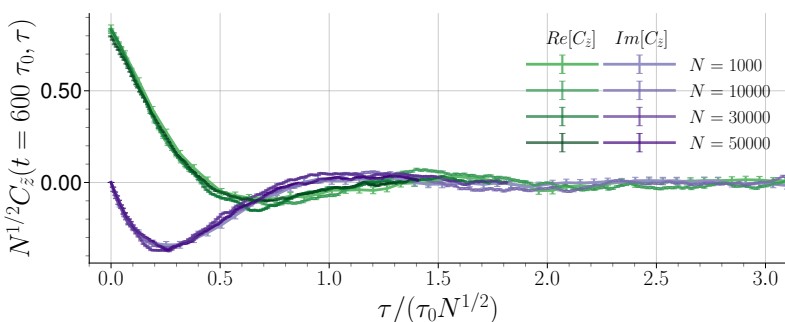

Figure 12: Hopf critical exponent: $\beta\lambda_+ = 1.0, \quad \beta\lambda_- = 1.7$. The time axis and auto-correlation are scaled according to Eq. 35 to show collapsing into a single function.

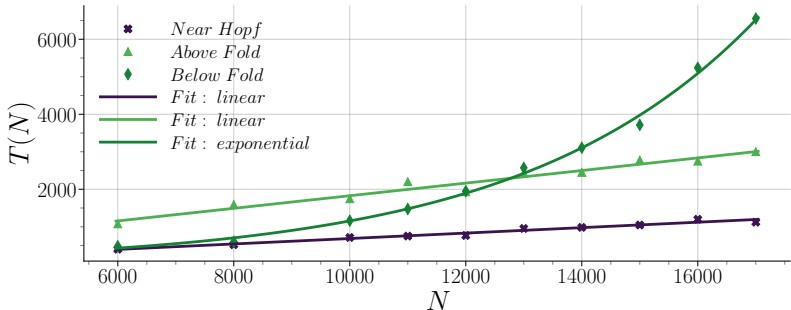

Figure 13: Characteristic decay times of autocorrelation near the Hopf line and above the Fold line grow linearly with $N$, following predictions of Eqs. 37 and 48. Below the Fold line, the decay time grows exponentially with $N$, following Eq. 50. Errors in the data are too small to be visible and have been omitted for clarity. The fits for all three regimes indicate strong statistical agreement between the data and the respective fitted models, with $R^2$ values of 0.969 (near Hopf), 0.980 (above Fold), and 0.993 (below Fold).

## 7 Conclusions

Several biological processes evolve through multi-step sequential transitions. Hematopoiesis, for instance, is a multi-step cascade that starts with stem cells and progresses through oligopotent and lineage-committed progenitors. Similarly, central pattern generators are neural circuits producing rhythmic or periodic functions such as breathing or walking. Another example is the cell cycle, which consists of a finely-tuned sequence of cellular phases. From a theoretical perspective, developing and understanding effective models that can address questions related to these biological sequential transitions is important. For instance, are these transitions controlled by intrinsic or extrinsic factors? What is the role of stochasticity, and how does it scale with the number of involved components? Are there critical regions that separate phases of different behaviors and exhibit some scale invariance properties? Are there critical regions of the phase space with enhanced sensitivity to external perturbations?

The two-memory non-reciprocal Hopfield model studied here addresses many of the above questions. Switching is encoded through non-reciprocal interactions that modify Hebbian coupling. In this $N$-body system, we explore the effects of the number of components, $N$, and noise. We found that two distinct regions of critical behavior emerge at the interface of different dynamical phases. We identified and studied these regions, which correspond to Hopf bifurcations and fold bifurcations. Previous studies have explored the hypothesis that some biological systems operate at Hopf bifurcation criticality. However, behaviors near the fold line

could explain other biological phenomena involving state switching. The dynamic scaling behavior, marked by different critical exponents $\zeta$ in the autocorrelation function, suggests these two regimes are qualitatively distinct. Furthermore, we showed that sensitivity to external signals varies significantly. Specifically, in the Hopf bifurcation line, the system is sensitive to perturbations resonant with the limit cycle frequency. In contrast, perturbations to a system in the fold line do not induce sustained limit cycles but enable controlled state switching. The time required to respond to perturbations also differs, scaling faster in the fold line than in the Hopf line.

The model studied here can be generalized to more than two patterns. For a system encoding $p$ patterns, the $N$ spins partition into $2^p - 1$ subnetworks, analogous to the division into similarity ($S$) and differential ($D$) spins introduced earlier. For instance, in Ref. [14], four patterns were used to extend the $Z_2 \times Z_2$ ($C_4$) model considered here to the $C_8$ symmetry case. A modification of the interaction using a Moore-Penrose pseudoinverse matrix of spins and patterns [39] was also used in that paper to reduce errors due to correlation among the memory patterns. However, for larger $p$, the model's ability to recover sequences of patterns quickly diminishes [3]. One way to address this limitation involves introducing a delay in the switching term [40], which could be realized through a modulation of the interaction, as recently explored by Herron et al. [41]. Hopfield networks do not need to be complete networks for memory retrieval. For instance, in random asymmetric networks, memory retrieval is preserved when the average network connectivity is above a critical value [42]. This property can be exploited to integrate the models with additional biological information. For example, in Ref. [25], the wiring of gene regulatory networks was combined with the memory retrieval property of the Hopfield model to identify bottleneck genes more susceptible to cell state switching. Another exciting extension involves defining branching points for memory patterns. Instead of cycles or fixed points, one can represent dynamics in which a memory pattern $\xi_1$ can transition into $\xi_2$ or $\xi_3$ patterns. This can be implemented by adding a random switch in the Glauber dynamics that randomly chooses between $\xi_2$ and $\xi_3$ in the dynamics. This approach was implemented in Ref. [13] to model the random switching between clonal states in disease progression. Finally, multilayer Hopfield networks offer an interesting generalization that brings several computational advantages, including the ability to disentangle complex patterns [43, 44]. In the context of biology, networks of Hopfield networks have been used to describe the interplay between gene-gene and cell-cell interactions, capturing spatial instabilities that lead to pattern formation in biological tissues [45].

While the present study has been motivated by biological questions, Hopfield networks with dilute memory patterns (i.e., $p < \log_2(N)$) have been explored in the presence of a transverse field on the $x$-axis, which renders the system quantum mechanical [46]. Although non-reciprocity in physical systems is less common than in biological settings, integrated photonics systems can be engineered to exhibit real space asymmetric coupling [47]. Non-reciprocity resulting from quantum mechanical effects in coupled parametric oscillators has also been recently demonstrated [47]. Studying these physical systems in critical regions near oscillatory instability could help understand the effects of noise and driving in truly out-of-equilibrium systems. Hopfield networks and their modern improvements [48] have also received renewed attention due to their connection to machine learning and artificial intelligence. For instance, new message-passing algorithms for Restricted Boltzmann Machines (RBM) have been proposed based on the mapping of Hopfield networks to RBM by a Hubbard-Stratonovich Gaussian transformation [49, 50]. Moreover, Hopfield networks have been suggested as a better alternative to the attention mechanism used in transformers [51]. Since the attention mechanism is the key innovation of the transformer architecture [52], a fundamental understanding of the properties of symmetric and asymmetric Hopfield neural networks could suggest more powerful architectures for AI applications.

# Acknowledgments

The content is solely the responsibility of the authors and does not necessarily represent the official views of the National Institutes of Health.

**Funding information**   Research reported in this publication was supported by the National Institute Of General Medical Sciences of the National Institutes of Health under Award Number R35GM149261. M.M. acknowledges support from the National Science Foundation under the NSF CAREER Award (DMR-2142866) as well as the NSF grant PHY-2112893.

**Code availability**   The code developed for this manuscript is available at: https://github.com/shuyue13/non-reciprocal-Hopfield and released on Zenodo: https://doi.org/10.5281/zenodo.16503491.

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
