# Peer review of "Critical Dynamics and Cyclic Memory Retrieval in Non-reciprocal Hopfield Networks"

_SciPost Physics, doi:SciPost Phys. 19, 100 (2025)_

## Round 1 · Referee Report · Anonymous (Referee 1) · 2025-4-1

Strengths

-it addresses Hopfield nets (with just a couple of patterns) via dynamical systems rather than statistical mechanical techniques, acting as a bridge among two well established disciplines.

-it paints a clear and coherent scenario for the network under study, in particular its dynamics is investigated in great detail.

-it constistutes a simple and transparent example of the rich behavior hidden in these Hebbian networks

-the language used to write the paper is a welcome tradeoff between intuitive explanations and mathematical formality

Weaknesses

-it focuses solely on two stored patterns.

Report

Please read the attached report.

Requested changes

Please read the attached report.

Attachment

Recommendation

Ask for minor revision

  • validity: good
  • significance: good
  • originality: good
  • clarity: good
  • formatting: good
  • grammar: good

Author:  Carlo Piermarocchi  on 2025-05-05  [id 5444]

(in reply to Report 1 on 2025-04-01)
Category:
answer to question

We are glad the referee found out manuscript interesting and well written and we thank them for the insightful comments and questions. Point by point answers are below.

Referee

Yet I did not understand, if I focus on the last two plots (the last row, with plots “e” and “f”) what is the center (m_1=m_2=0)? Furthermore, you have four sinks because you store the two patterns and the two gauge-related patterns? (i.e. \xi^1 and -\xi^1 for pattern 1)?

Answer The eigenvalues of the Jacobian matrix $A$ at $m_1=m_2=0$ are $\beta \lambda_+ -1 \pm i \beta \lambda_-$ so in the regime of parameters in plots “e” and “f” both eigenvalues have a positive real part, which makes the fixed point unstable (source node). We modified the caption to indicate that in “e” and “f” the central point becomes an unstable fixed point. The referee is correct. The patterns $\xi^i$ and $-\xi^i$ are equivalent so the four sinks reflect that.

Referee

In sec. 3.1 I do not entirely understand why in eq. 25 the r.h.s is a free energy and not a standard energy function. […] the same problem is in eq. (33) where I recognize an energy but barely a free energy...

Answer We originally wrote “free energy” because the phase diagram is discussed in terms of the $\beta\lambda_\pm$ parameters, which depend on the temperature through $\beta$. However, for the sake of the discussion in these sections, it is clearer to talk about effective Hamiltonians, as the referee suggests. We added the “effective” to emphasize that the original system with non-reciprocal interaction cannot be described by a Hamiltonian in the traditional sense. We changed $F$ to $H$ in Eqs. 25 and 33.

Referee

In Sec. 3.3 the question of the Goldstone mode is interesting but subtle (I already seen this in Andrea Cavagna’s papers): I would add a citation to a paper that the Authors think relevant for understanding for the general reader not aware of a Goldstone mode...

Answer We added a textbook reference on the Goldstone theorem and cited two recent works where similar remarks have been made in the context of dynamic limit cycles.

Referee

-After eq. 61, the Authors cite (using their bibliography) [36,37] to highlight research on networks without self-interactions but those papers where on a slighlty different problem: Personnaz and coworkers were investigating unlearning protocols in Hebbian nets, while Kanter and Sompolinsky worked out the statistical mechanical version of the Kohonen net, yet these two papers are deeply linked as the (correct) unlearning scheme for the Hopfield network allows the model to collapse to the Kanter-Sompolinsky one as explained in […]Further, along the same line, I also point out that both the research groups on neural nets in Rome and Tokyo are inspecting very similar research lines, see e.g. […]

Answer We really enjoyed reading the papers suggested by the referee on the role of self-interaction in unlearning and its connection to Kohonen networks. The discussion of self-interaction in the $p/N \gg 1$ limit in Saad’s paper was also interesting. However, since our paper focuses on two memory patterns, which we assume to be orthogonal, we feel that including a discussion of this point in that section would unnecessarily complicate the presentation. Therefore, we have decided to remove the misleading comment. In contrast, we have cited some of the other papers suggested by the referee in the conclusion section to highlight the extension to networks of Hopfield networks and their relevance to biology.

Referee

Also, as a last point regarding the bibliography, I think that a very early PNAS by Amit -where the idea of coupling \xi^{\mu} to \xi^{\mu+1} was introduced- is missing […]

Answer The content of the PNAS paper is included in Amit’s book, which we already cited in the introduction. However, for completeness, we now also cite the original PNAS paper.

Referee

Finally, a question I’d like to ask is about the stability of the painted picture when the number of patterns is minimally increased […].

Answer We have attempted to extend the approach to the three-memory case. As in the case of differential and similarity subnetworks, the three-memory network can also be decomposed into subnetworks of equivalent sites. While mean-field equations analogous to those presented in this paper can be derived, we have not found an elegant framework similar to the two-memory case. We plan to explore this further in future work.

Author:  Carlo Piermarocchi  on 2025-05-05  [id 5443]

(in reply to Report 1 on 2025-04-01)

We are glad the referee found our manuscript interesting and well written, and we thank them for their insightful comments and questions. The point-by-point answers are below.

Referee:

Yet I did not understand, if I focus on the last two plots (the last row, with plots “e” and “f”) what is the center (m_1=m_2=0)? Furthermore, you have four sinks because you store the two patterns and the two gauge-related patterns? (i.e. \xi^1 and -\xi^1 for pattern 1)?

Answer The eigenvalues of the Jacobian matrix $A$ at $m_1=m_2=0$ are $\beta \lambda_+ -1 \pm
i \beta \lambda_- $
so in the regime of parameters in plots “e” and “f” both eigenvalues have a positive real part, which makes the fixed point unstable (source node). We modified the caption to indicate that in "d" , “e” and “f” the central point becomes an unstable fixed point. The referee is correct, $\xi^i$ and $-\xi^i$ are equivalent patterns, so the four sinks reflect that.

Referee

In sec. 3.1 I do not entirely understand why in eq. 25 the r.h.s is a free energy and not a standard energy function. [...] the same problem is in eq. (33) where I recognize an energy but barely a free energy...

Answer We originally wrote “free energy” because the phase diagram is discussed in terms of the $\beta\lambda_\pm$ parameters, which depend on the temperature through $\beta$. However, for the sake of the discussion in these sections, it is clearer to talk about effective Hamiltonians, as the referee suggests. We added the “effective” to emphasize that the original system with non- reciprocal interaction cannot be described by a Hamiltonian in the traditional sense. We changed $F$ to $H$ in Eqs. 25 and 33.

Referee

In Sec. 3.3 the question of the Goldstone mode is interesting but subtle (I already seen this in Andrea Cavagna’s papers): I would add a citation to a paper that the Authors think relevant for understanding for the general reader not aware of a Goldstone mode...

Answer We added a textbook reference on the Goldstone theorem and cited two recent works where similar remarks have been made in the context of dynamic limit cycles.

Referee

-After eq. 61, the Authors cite (using their bibliography) [36,37] to highlight research on networks without self-interactions but those papers where on a slighlty different problem: Personnaz and coworkers were investigating unlearning protocols in Hebbian nets, while Kanter and Sompolinsky worked out the statistical mechanical version of the Kohonen net, yet these two papers are deeply linked as the (correct) unlearning scheme for the Hopfield network allows the model to collapse to the Kanter-Sompolinsky one as explained in [...]Further, along the same line, I also point out that both the research groups on neural nets in Rome and Tokyo are inspecting very similar research lines, see e.g. [...]

Answer We really enjoyed reading the papers suggested by the referee on the role of self- interaction in unlearning and its connection to Kohonen networks. The discussion of self- interaction in the p/N≫1 limit in Saad’s paper was also interesting. However, since our paper focuses on two memory patterns, which we assume to be orthogonal, we feel that including a discussion of this point in that section would unnecessarily complicate the presentation. Therefore, we have decided to remove the misleading comment. In contrast, we have cited some of the other papers suggested by the referee in the conclusion section to highlight the extension to networks of Hopfield networks and their relevance to biology.

Referee

Also, as a last point regarding the bibliography, I think that a very early PNAS by Amit -where the idea of coupling \xi^{\mu} to \xi^{\mu+1} was introduced- is missing [...]

Answer The content of the PNAS paper is included in Amit’s book, which we already cited in the introduction. However, for completeness, we now also cite the original PNAS paper.

Referee

Finally, a question I’d like to ask is about the stability of the painted picture when the number of patterns is minimally increased [...].

Answer We have attempted to extend the approach to the three-memory case. As in the case of differential and similarity subnetworks, the three-memory network can also be decomposed into subnetworks of equivalent sites. While mean-field equations analogous to those presented in this paper can be derived, we have not found an elegant framework similar to the two-memory case. We plan to explore this further in future work.

---

## Round 1 · Referee Report · Anonymous (Referee 1) · 2025-5-9

Strengths

The same of the previous version (nothing has substantially changed as there were just minor revisions to be implemented).

Weaknesses

The same of the previous version (nothing has substantially changed as there were just minor revisions to be implemented).

Report

I do believe that this paper, in its present revised form, should appear on this Journal as it provides and deepens a new link between the Hopfield model as a dynamical system and the Hopfield model as a statistical mechanical reference. This bridge could give rise to several follow-up work as it has just been proposed but there is a long way to go before such an exploration is over.

Furthermore, it also shows how to use know-how stemming from neural networks into a broader scenario of general dynamical systems thus making the Hopfield model a candidate to account also other emerging properties beyond the computational capabilities of neural networks.

Nothing prevents me to suggest acceptance of this very nice paper in its present form.

Recommendation

Publish (easily meets expectations and criteria for this Journal; among top 50%)

---

## Round 1 · Referee Report · Anonymous (Referee 2) · 2025-9-9

Disclosure of Generative AI use

The referee discloses that the following generative AI tools have been used in the preparation of this report:

ChatGPT to polish the english style

Strengths

1. The exploration of non-reciprocal Hopfield networks via dynamical systems is original and important. The identification of distinct scaling exponents at Hopf and fold bifurcations provides new insights into dynamical universality classes relevant to out-of-equilibrium systems.
2. The combination of analytical approaches with numerical methods makes the results robust and comprehensive.
3. The manuscript is very well organised. Each section logically develops from the previous one, with clear explanations of both the physical intuition and technical derivations.
4. The results not only enrich our understanding of dynamical phase transitions in neural networks but may also inspire applications in systems biology and machine learning, particularly in designing architectures that exploit cyclic dynamics or critical sensitivity.

Weaknesses

It only considers two stored patterns.

Report

The manuscript "Critical Dynamics and Cyclic Memory Retrieval in Non-reciprocal Hopfield Networks" by S. Xue et al presents a thorough and timely study of Hopfield networks with non-reciprocal couplings, focusing on the emergence of cyclic memory retrieval and the associated critical dynamics. The authors analyse the phase diagram, identify Hopf and fold bifurcation lines, and characterise the scaling behaviour of autocorrelation functions and response times near these critical regions. Analytical predictions are corroborated with a combination of Master Equation approaches, Liouvillian diagonalization, and Glauber dynamics simulations. The work also draws connections to biological systems, where cyclic instabilities and critical sensitivity to perturbations are highly relevant. The authors have properly and convincingly replied to a number of points raised by the first referee.

Requested changes

None

Recommendation

Publish (easily meets expectations and criteria for this Journal; among top 50%)

---

## Editorial Decision

published